# Requirements for Crafting Virtual Network Packet Captures

Daniel Spiekermann [1] and Jörg Keller [2,*]

1 Polizeiakademie Niedersachsen, 31582 Nienburg, Germany; daniel.spiekermann@polizei.niedersachsen.de
2 Faculty of Mathematics and Computer Science, FernUniversität in Hagen, 58084 Hagen, Germany
* Correspondence: joerg.keller@fernuni-hagen.de

**Abstract:** Currently, network environments are complex infrastructures with different levels of security, isolation and permissions. The management of these networks is a complex task, faced with different issues such as adversarial attacks, user demands, virtualisation layers, secure access and performance optimisation. In addition to this, forensic readiness is a demanded target. To cover all these aspects, network packet captures are used to train new staff, evaluate new security features and improve existing implementations. Because of this, realistic network packet captures are needed that cover all appearing aspects of the network environment. Packet generators are used to create network traffic, simulating real network environments. There are different network packet generators available, but there is no valid rule set defining the requirements targeting packet generators. The manual creation of such network traces is a time-consuming and error-prone task, and the inherent behaviour of virtual networks eradicates a straight-forward automation of trace generation in comparison to common networks. Hence, we analyse relevant conditions of modern virtualised networks and define relevant requirements for a valid packet generation and transformation process. From this, we derive recommendations for the implementation of packet generators that provide valid and correct packet captures for use with virtual networks.

**Keywords:** virtual networks; packet generation; packet transformation; network forensic investigation





## 1. Introduction

Virtual machines, virtual storage and virtual networks are the fundamentals of modern infrastructures, providing highly flexible applications, services and environments such as Software-as-a-Service (SaaS), Platform-as-a-Service (PaaS) and Infrastructure-as-a-Service (IaaS). Cloud environments as well as modern company networks use virtualisation to provide dynamic, secure and highly customisable environments to internal and external customers.

These environments have to cover various, partly conflicting challenges. Inter-networking and connections to internal and external networks are as important as the isolation and secure separation of different networks categories such as customers, management and backup. The implementation of network virtualisation overlay (NVO) is a common technique to create flexible overlay networks on top of common and hardware-based underlay networks. The software-based overlay networks use different virtual network protocols such as VXLAN [1], GENEVE or NVGRE [2] to create a flexible network structure for the different tenants. These protocols facilitate the separation of different networks on top of a single and independent underlay network by encapsulating a given network packet with new protocol information as shown in Figure 1.

The use of encapsulating protocols provides various benefits to the providers. The encapsulation provides flexibility to static networks by adding new network traffic information to a given network packet. The idea of encapsulating network protocols has been known for a long time and is not limited to Ethernet or IP [3]. In addition to this, encapsulating protocols are not limited to local area networks (LAN), even in wide area networks (WAN) protocols such as Point-to-Point Protocol (PPP) [4] or ATM (Asynchronous Transfer Mode) use encapsulation.

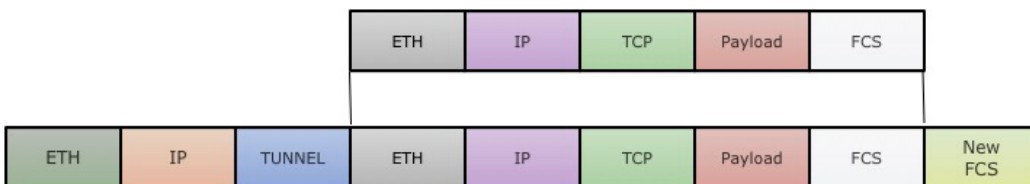

**Figure 1.** Network packet encapsulation.

In modern networks, protocols such as VXLAN use a special field in the header that identifies an isolated subnet under the administration of a single customer or for a special purpose. In this way, subnets belonging to the same entity can be spread all over the physical infrastructure. If a host wants to communicate with a host located in the same subnet but on a different compute host, the network packet is encapsulated.

The administration of these networks differs from the administration of pure hardware-based environments. Now, additional attack vectors such as denial-of-service attacks against network controllers [5], insider attacks against virtualised devices [6] or software crashes and buffer overflows of the software implementations are gaining increased importance.

To manage the different challenges, modern techniques such as machine learning-based intrusion detection or network packet classification with deep learning are implemented, but tasks such as prediction occur as well. To train the involved algorithms, a huge amount of network traffic is needed. A common approach is the use of available datasets, but this information differs sometimes from the given network environment. To obtain proper network traffic, [7] describes six different techniques to generate network packets or traffic collection:

1.  Use of real networks,
2.  Creation of a honeynet,
3.  Use of a network simulator,
4.  Existing traffic dumps,
5.  Use of network traffic generators,
6.  Combining the aforementioned techniques.

The use of real network traffic seems to be a common possibility to get relevant network traffic, but this traffic lacks in attack-based network traffic or malicious packets. The creation of a honeynet as well as the network simulator is easier in virtual environments as in hardware-based environments but still a complex task. The process of implementing such an environment is done in various steps. In [8], the authors define significant planning as the first step, along with discussions of trained personnel to preserve the realistic scenario. The technical implementation is based on the installation and configuration of VMs running on compute nodes as well as the wanted network structure. If dynamic aspects such as VM migration have to be covered, a reconfiguration of the entire environment is necessary.

While a huge number of datasets of common networks such as CICDDoS2019 [9], CSE-CIC-IDS 2018 [10] and INSecS-DCSl [11] exist, to our knowledge, there is no publicly available data set with different virtual protocols such as VXLAN or GENEVE.

The use of network traffic generators provides various benefits to the provider. By defining coherent parameters, a packet generator is able to create proper network traffic which fits the involved network. In [12], the authors define packet generation as

*the result of time-stamped series of packets arriving and departing from particular network interfaces with realistic values.*

We define network packet generation as the synthetic creation of network packets regarding different parameters. These parameters depend on the purpose of the creation process and cover various fields such as the testing of new implementations [13], troubleshooting [14], education and training [15,16] or advanced aspects such as network forensic investigation [17,18].

However, ad hoc traffic generation is cumbersome and error-prone and normally does not take virtualised networks into account. Packet generators are used in various

fields of network management, security and performance optimisation. In [19], the authors describe the use of different network packet generators to implement training platforms for teaching defence techniques against denial of service attacks. With *Encapcap* [20], we have presented a first attempt to transform network traces so that they contain virtualised network traffic for specific scenarios. In the present research, we investigate systematically the requirements to be met by packet generators for virtualised networks and thus to see which direction to evolve *Encapcap* in for use with more general scenarios. We also present some use cases that illustrate the necessity of these requirements and demonstrate the use of *Encapcap* in application scenarios. While those use cases comprise the use of test sets for virtual networks generated by *Encapcap*, our goal is not to present a particular test set but the possibility to generate it.

Our main research contributions are as follows:

- An analysis of relevant features in traffic from virtualised networks and subsequently the derivation of a set of requirements, which is as complete as possible, as a baseline for a packet generation process targeted towards virtual environments;
- The extension of the *Encapcap* tool to meet the above requirements;
- The presentation and evaluation of different use cases with the help of (extended) *Encapcap*.

The remainder of this article is structured as follows. Section 2 discusses related work in the fields of virtual networks and packet capture generation. In Section 3, we define pertinent requirements for a valid packet generation process targeted towards virtualised networks. Section 4 presents and analyses use cases, while Section 5 presents conclusions and an outlook to future work.

## 2. Related Work

The creation of network packets is discussed in various research works. In [21], the authors propose Moongen as a high-speed packet generator able to saturate a 10GBit/s connection. Pktgen [22] is another software-based packet generator that aims to cover high-speed communication. In addition to this, different hardware-based generators such as [23] exist.

Encapsulating protocols are able to tunnel given protocol information to transfer this information without any further interaction from a given point in a network to its intended destination. This is done for security reasons such as IPSec and Encapsulating Security Payload (ESP) [24], virtual private networks [25] or virtual environments [26]. Virtual environments such as software-defined networks (SDN), Docker [27] or Kubernetes [28,29] are heavily based on encapsulating protocols [30]. Network packet captures freeze the transferred network traffic and, depending on the captured traffic, provide additional opportunities for training, learning and testing.

The generation of own packet captures is done for various reasons such as testing of new network protocols, fuzzing, IoT design [31] or security implementations [32]. Because of this, [33] differentiates application-level, flow-level and packet-level traffic generators.

A common technique is the replay of the network packets inside a virtual environment with tools such as tcpreplay, TCPivo [34] or OFRewind [35]. The tools provide great flexibility in repeating captured network traces but focus on the implementation or evaluation of high-performance networks. In contrast to these, tools with a forensic purpose such as [36] focus on making the replay as accurately as possible. When crafting network packets with encapsulation protocols as used in SDN, only *Encapcap* [20] is available. The definition of requirements typically depends on the intended purpose of the generator; e.g., in [37], the authors describe bandwidth, accuracy and precision as relevant requirements for network packet generators, but this research focuses on the live injection of the crafted network packets into a network or network analysis tool. In [38], the authors define five requirements supporting the reproducible traffic generation. In [39], the authors define seven requirements in virtual networks, but they focus on benchmarking and measurement.

## 3. Requirements

The synthetic creation of valid packet captures, typically stored as pcap files, (https://en.wikipedia.org/wiki/Pcap, accessed on 27 June 2022) is complex and faced with challenges. Errors, misconfigurations or missing considerations of existing standards might lead to incorrect and futile capture files. To facilitate valid and correct capture files, we present the following requirements, which help to overcome these challenges. Thus, we also give recommendations for generators with respect to these requirements.

To ensure a systematic process and achieve a list of requirements that is as complete as possible, we took software testing as an example and role model [40]. In addition, we used our multi-year experience in this field. Yet, we are aware that—as always—such a list of requirements remains a best effort approach. Software testing has to ensure the *correctness* of the different algorithms implemented in the software, e.g., in the form of unit tests. IN addition, the different parts must work together and must be *compatible* with the outside world, e.g., checked by integration tests. Testing needs to cover as many program paths as possible, i.e., to show *flexibility* to address different circumstances, and need to rerun when circumstances change, e.g., in regression testing; i.e., they must be *adaptable*. Tests must be *reproducible*, but, e.g., in black-box testing, *randomisation* is needed. Tests should not only consider functionality but must also address *performance*. Especially for embedded and real-time systems, also predictable and acceptable temporal behaviour like response time must be checked, which we cover under the term *precision*. Our last requirement refers to appropriate use case design: a generator must be *aware* of virtual networks to address their specific features.

### 3.1. Correctness

The correctness of data is a critical aspect in all digital investigation [41] as well as testing and development. Thus, the correctness of synthetically created network packets is crucial to the overall process. Incorrect or partial network packet captures are invalid and should not be used in the further process of analysis. Missing or distorted network packets have an unpredictable effect on the subsequent steps, which might result in incorrect or misleading results. Common errors are missing values, incomplete packet flows and wrong check-sums resulting in new errors when investigating or analysing corrupt data. Especially in the field of machine learning, the use of valid and appropriate test data is crucial for the effective detection of unwanted traffic. Neglecting the correctness is possible if payload content does not matter and only header information is analysed [42].

### 3.2. RFC Compatibility

The process of generating network packets is mostly done for a specific purpose. As mentioned in the previous part, the correctness of the packets is a crucial aspect. In addition to this, the compatibility to the most recent Request-for-Comments (RFC) is important. These standards define relevant aspects such as protocol structures, valid and invalid entries, default values and recommendations for processing the information. If these standards or parts of them get ignored, different follow-up errors might arise. These errors might detract from the intended purpose of the packet capture. Hence, it is crucial that the generated network packets consider the valid specifications. This is a time-consuming part of the generation, because the different specifications and their expansions have to be analysed in detail. Some specifications depend on each other, expand older versions and add new functions. As an example, GRE as defined in RFC 2784 [43] does not provide the relevant header fields for NVGRE, but the extension of GRE defined in RFC 2890 [44] adds two optional header fields to the original header with a length of 32 bits each. NVGRE uses 24 bits to store the virtual subnet ID (VSID). The remaining 8 bits are used as a FlowID, which provides per-flow entropy for flows in the same VSID.

### 3.3. Flexibility

Packet captures of hardware-based networks provide mostly static information, as the number of changes inside this environment is very low. Connected systems such as servers or computers do not change in these environments, which results in a predictable network design. A major benefit of virtual networks and virtual environments is the flexibility and dynamic of the infrastructure. Typically, packet captures of virtual environments contain a huge number of changing network information such as IP addresses, protocols and involved hosts. This is a result of VMs that are started and stopped inside the environment, or suspended systems copied to different locations inside a user network. A packet generation process in this environment should provide the same flexibility as the environment; otherwise, the generation process is too static for a modern environment. As a result, the generator should be able to add additional systems into the capture file or change information of existing systems.

### 3.4. Adaptability

In all fields of information security or digital investigation, the use of valuable test data is a perpetual challenge [8]. In contrast to traditional hardware-based environments, it is not applicable to use purchased second-hand hardware to collect different user data [45]. Unfortunately, network data are volatile and not stored on hardware after the transmission; thus, purchasing old hardware is not a reasonable approach. However, the need for different scenarios based on various data is crucial. In [46], the authors define the need for realistic training as follows: "In order to prevent or detect cyber-attacks, people soon realised that the best way is practising hands-on training, where trainees work in a testing environment that mimics real-life situations." Without this variety of scenarios, the effect of training is limited and does not guarantee an ongoing improvement of the involved personnel [47]. With different scenarios and various levels of difficulty, based on flexible generated network captures, the overall acceptance of training and testing can be improved. The packet generator has to create adaptable capture files based on given files. The adaptability can be reached by a useful software design like a graphical user interface or a parameter-based command line interface.

### 3.5. Reproducability

In addition to the correctness of the created packet captures, it is necessary that the results of packet transformation are repeatable and resilient. The reproducability is a critical requirement of digital investigations [48,49] and ensures the equality of two (or more) generated packet captures based on the same original file and that are manipulated in the same manner. The generation process should be automatised and perform the same steps for every run. Use of parameter sets to characterise and store all relevant parameter settings for a particular generation run in one place can be of help here.

### 3.6. Randomisation

On the other hand, the generator should be able to create randomised values, which change different packet parameters if needed. By using pseudo-random number generators of good quality [50] and by recording the seed value as part of the parameter set, the requirements of reproducability and randomisation can be fulfilled together.

### 3.7. Performance

The necessary speed of the generation process depends on the intended purpose. If the generation process injects the packets into a real network in real time, the performance of the process should touch 10 GBit/s [22]. A limiting factor of a generation process is the deployed hardware, but the design of the software heavily affects the overall performance. In addition to this, the memory allocation of the creation process should be suitable for the created capture file.

*3.8. Precision*

A packet generator has to create precise packet captures with defined gaps or intervals between network packets, which results in accurate packet captures [21,51]. In particular, the pcapng-format is able to manage network packets with a timestamp of 64-bit accuracy, so a packet generator should be able to create correct timestamps in a packet.

However, a common issue of different network packet generators is a varying rate of packet transmission. Software-based solutions depend on the CPU load. If a generator such as *tcpreplay* is used multiple times to replay a given capture file, the time between sent network packets might differ from run to run [52]. If there are competitive processes on a single-core system, the replay process is not eligible to use the CPU core in time.

*3.9. Awareness*

Virtual networks differ from physical networks in various aspects which have to be handled during the transformation. Techniques such as migration or on-demand changes in the infrastructure, often initiated by a user, are common in virtual environments. A packet generator for virtual environments has to be aware of these situations and has to be able to create valid scenarios based on a given packet capture.

Another event that can occur in virtual networks is the change of internal network packet information and addresses such as the IP range of a user network. A customer is able to change the assigned network and might change IP addresses, routes to some destinations, packet forwarding, firewalling or add additional NFV devices such as VPN gateways or load balancers. All these changes have effects on the network and have to be managed by a packet generator, i.e., by changing internal network information of the transferred network packets.

A further difference to traditional hardware-based networks is the possibility to change protocols on the fly. This protocol swap is done to implement additional features scgh as TCP-offloading [53]. A packet generator should be able to manage such a protocol change by providing the ability of swapping from one protocol to another, which results in different NVO information such as protocol types or port numbers. A packet generator has to analyse and change the necessary information on the fly to create valid and usable capture files.

*3.10. Extending Encapcap*

Given the systematic analysis of requirements above, we analysed how far our own tool *Encapcap* [20] already fulfills these requirements and where it can be improved.

The analysis resulted in the following extensions. The first release of *Encapcap* as discussed in [20] was able to create virtual network packet captures from a given capture file containing arbitrary network packets. A check for correctness and RFC compatibilitiy of the initial trace has been added. When started with a list of various parameters *Encapcap* is able to provide flexible and adaptable but also reproducible packet captures. A special parameter *--rand* creates randomised values, which prevents the correct reproducability of a created packet capture. As a consequence, the use of *--rand* has been extended to include a mandatory seed value in the current release of *Encapcap*. The extension of *Encapcap* provides a tool that covers all aforementioned requirements with the exception of performance. *Encapcap* is implemented in Python3, so the performance of the creation process is limited due to the interpreting speed of the Python language. Implementing a packet generator in a compiling language such as *C* will heavily improve the performance of *Encapcap* [54] and is thus targeted as future work.

**4. Use Cases**

In this section, we list three different use cases, which illustrate the proposed requirements. Based on the scenario, the requirements have different relevance. Not every requirement is critical for every use case, but with three use cases, we illustrate that the complete set of requirements is necessary; e.g., the performance as discussed in Section 3.7

is mostly irrelevant, when the need for having adapted network captures is not urgent or the process of creating network packets does not have to be conducted in a short time. On the other hand, a flexible adaptation of the resulting network capture as discussed in Section 3.3 is important for network security. This area has to manage different network traffic, and various attacks use an unusual combination of protocol header fields [55] or network internal aspects such as fragmentation [56]. In addition to this, various network based covert channels use different header fields to exfiltrate the intended information [57,58].

## 4.1. Training

The implementation of new network protocols might result in new or unknown errors, e. g., in the case of misconfiguration of involved systems or misinterpretation of specifications. If a problem occurs, a common technique is network packet analysis [59]. Such an analysis aims to detect irregular values in packet headers or the incorrect implementation of given standards. Packet captures with pre-defined values related to a given network help to train network engineers to learn new protocols in order to prevent the aforementioned issues. As discussed in Section 3.2, the creation of the requested packet captures needs to consider valid specifications; otherwise, emerging errors might hamper the intended analysis. Figure 2 shows such an intentionally misconfigured capture file that violates the addressing scheme for MAC-addresses defined in [60]. Such a misconfiguration might confuse the user of the tool and distract from other, more relevant issues or the intended purpose. Thus, such misconfiguration should not occur within training sets for network engineers. So, considering specifications and creating correct captures are critical requirements for network packet generations. Otherwise, there would be a need for additional checks after the creation of the transformed dataset before using it in an actual training session. The use of *Encapcap* allowed us to create traces covering 1 hour of packets for training faster.

```
> Frame 1: 98 bytes on wire (784 bits), 98 bytes captured (784 bits)
> Ethernet II, Src: 91:0a:52:58:00:04 (91:0a:52:58:00:04), Dst: 73:bc:ac:dd:04:5c (73:bc:ac:dd:04:5c)
  > Destination: 73:bc:ac:dd:04:5c (73:bc:ac:dd:04:5c)
  v Source: 91:0a:52:58:00:04 (91:0a:52:58:00:04)
    > [Expert Info (Warning/Protocol): Source MAC must not be a group address: IEEE 802.3-2002, Section 3.2.3(b)]
      Address: 91:0a:52:58:00:04 (91:0a:52:58:00:04)
      .... ..0. .... .... .... .... = LG bit: Globally unique address (factory default)
      .... ...1 .... .... .... .... = IG bit: Group address (multicast/broadcast)
    Type: IPv4 (0x0800)
> Internet Protocol Version 4, Src: 192.168.10.1 (192.168.10.1), Dst: 192.168.10.198 (192.168.10.198)
> Internet Control Message Protocol
```

**Figure 2.** Example of incompatibility to RFC.

Other requirements such as adaptability, as discussed in Section 3.4, or reproducability, discussed in Section 3.5, help training scenarios to create a clean training environment that provides an identical basis for different training scenarios

## 4.2. Testing

The network is a fundamental and crucial element in modern environments. The high-speed connection of involved systems as well as the correct transport of the information is one of the most relevant tasks. Bringing new devices into the network has to be tested in order to minimise the risk of unwanted results. Such devices, hardware-based as well as virtual, change the internal packet flow, which results in a different behaviour. So, testing a new device in a real-world scenario is necessary in modern infrastructures, but most of the environments do not provide such an opportunity. Creating network packets with real-world information in real time supports the testing and implementation of new devices. The generated packets, either by a tool such as *Encapcap* or by a different tool that is able to provide virtual network packet captures, can be replayed or sent in real time into a given network device, and the resulting behaviour can be analysed in detail in a simulated network environment to ensure the intended processing. A slow or dissimilar packet flow might not produce the same behaviour as high-speed packet transfers with valid network data, so the precision and performance as well as the awareness are crucial requirements in this use case. We evaluated this process by replaying a generated packet capture into a

firewall implementation to analyze the forwarding process of the device. By this, we were able to detect minor misconfigurations in the complex ruleset of the firewall. Without such a testing process, these errors would have persisted into production mode. As a result, this testing opportunity is similar to gray box testing [61].

### 4.3. Security

Network security and protection is relevant in modern environments. Due to the increasing speed and diversity of network packets, the detection of network-based attacks is a complex task. Preventing adversarial attacks is typically done with techniques such as machine learning, but the deployed algorithms need realistic and correct network packets to train and improve the models. Incorrect, static or incongruous information leads to wrong results and therefore useless models. So, training the models with appropriate information is crucial for a high level of security. As shown in [18], the shift from hardware-based to virtual networks impacts every involved network device that analyses the transferred network traffic. Whereas performance issues or a huge resource load might be detected in an easy way, the detection of misconfigured or incomplete rules for the protection of the network is critical but extremely difficult. Because of this, the process of implementing new network devices or new algorithms for the classification of network packets in virtual networks needs a valid baseline of well-known information. By using a known data-set of the traditional network and transferring this capture to a capture file with adapted information of the intended virtual network, the overall security benefits result in a clean and correct process of packet generation. The re-configuration of the firewall mentioned in Section 4.2 improved the security of a virtual network.

### 4.4. Relation to Requirements

The requirements defined in Section 3 cover the relevant aspects of the aforementioned realistic use cases. Table 1 summarises the listed use cases and their corresponding requirements. An *x* marks the need of the requirement for the given use case, whereas – defines a lower significance for the use case.

**Table 1.** Use case analysis.

| Requirements | Use Case 1 Training | Use Case 2 Testing | Use Case 3 Security |
|---|---|---|---|
| Correctness | x | x | x |
| RFC compatibility | x | - | - |
| Flexibility | x | - | x |
| Adaptability | x | x | x |
| Reproducability | x | x | x |
| Randomisation | - | - | x |
| Performance | - | x | - |
| Precision | x | x | - |
| Awareness | - | x | x |

Table 1 illustrates that not every requirement is critical for every use case, but that the complete set of requirements is necessary already for the three given use cases. Because of the variety of the different reasons for packet generation, a different emphasis in other scenarios is possible. On the other hand, Table 1 details that the correctness is the most relevant requirement when creating adapted virtual network packets. If the generation of virtual packet traces produces irregular or incorrect data, every subsequent step such as flow creation, feature selection, in-depth analysis or performance calculation will result in misleading data, errors or unwanted behaviour.

## 5. Conclusions

Virtual networks are on the rise, and hence packet traces of virtual networks are needed for a variety of use cases ranging from machine learning-based firewall configuration to realistic training sessions for network engineers.

The process of creating such a dataset for virtual networks is a time-consuming and error-prone task, which is faced with different issues. If the creation process is not fully applicable to the intended purpose, subsequent errors might be undetected. In this paper, we therefore derived, in a systematic manner and as completely as possible, a set of requirements for the creation of datasets in modern networks, with an eye on virtualisation techniques.

We have extended our own tool *Encapcap* [20] with the knowledge from the systematic analysis in this research to meet all requirements except performance. We have applied this tool in several use cases and have illustrated that the set of requirements is indeed necessary for successful application.

Future work comprises the re-implementation of *Encapcap* in a compiled programming language to improve performance as the last missing requirement.

**Author Contributions:** Conceptualization, D.S. and J.K.; methodology, D.S.; software, D.S.; validation, D.S.; formal analysis, D.S. and J.K.; investigation, D.S.; resources, D.S.; data curation, D.S.; writing—original draft preparation, D.S. and J.K.; writing—review and editing, D.S. and J.K.; visualization, D.S.; All authors have read and agreed to the published version of the manuscript.

**Funding:** This research received no external funding.

**Institutional Review Board Statement:** Not applicable.

**Informed Consent Statement:** Not applicable.

**Data Availability Statement:** Not applicable.

**Conflicts of Interest:** The authors declare no conflict of interest.

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
