# Peer review of "Requirements for Crafting Virtual Network Packet Captures"

_jcp, doi:10.3390/jcp2030026_

Round 1

Reviewer 1 Report

This paper investigate the requirements to be met by packet generators for virtualised networks. The process of creating such a dataset is a time-consuming and error-prone task, it is faced with different issues. The process is not fully applicable to the intended purpose, subsequent errors might be undetected.

It is a good work. I think this work can be accepted.

Author Response

Dear Editor, Dear Review,

please find our reply letter below.

Yours Sincerely, J. Keller

Reviewer 2 Report

The article is focused on generating virtual network packets.
One of the main benefits presented by authors in the article is the analysis of relevant variables in operation from the virtual network. The authors do not describe the obtained data in depth, I miss a more detailed description of the potential dataset (s). In this area, machine learning methods and deep learning are used to solve various problems, which, however, requires a sufficient amount of data. The authors state this fact, but they do not describe in depth which tasks in terms of knowledge acquisition are most often solved in this domain (prediction, classification, segmentation, ..). It would
be desirable to pay deeper attention to this issue, ie. what methods and to solve what problems, respectively tasks in terms of knowledge acquisition are used. The specific structure and nature of
the data predetermine the use of specific machine learning methods.
At the end of the article, the authors reiterate the requirement to create usable datasets for analysis purposes also through machine learning methods.

Author Response

(The authors gave the same response as above.)
